# The Relaxation Behavior of Water Confined in AOT-Based Reverse Micelles Under Temperature-Induced Clustering

**DOI:** 10.3390/ijms26157152

**Published:** 2025-07-24

**Authors:** Ivan V. Lunev, Alexander N. Turanov, Mariya A. Klimovitskaya, Artur A. Galiullin, Olga S. Zueva, Yuriy F. Zuev

**Affiliations:** 1Kazan Institute of Biochemistry and Biophysics, FRC Kazan Scientific Center, Russian Academy of Sciences, Lobachevsky Str. 2/31, 420111 Kazan, Russia; lounev75@mail.ru (I.V.L.); mklimovitskaya@mail.ru (M.A.K.); 2Physical Institute, Kazan Federal University, Kremlevskaya Str. 18, 420008 Kazan, Russia; www.ag95@mail.ru; 3Zavoisky Physical-Technical Institute, FRC Kazan Scientific Center, Russian Academy of Sciences, Sibirsky Tract 10/7, 420029 Kazan, Russia; sasha_turanov@rambler.ru; 4A.M. Butlerov Chemical Institute, Kazan Federal University, Kremlevskaya Str. 18, 420008 Kazan, Russia; 5Institute of Electric Power Engineering and Electronics, Kazan State Power Engineering University, Krasnoselskaya Str. 51, 420066 Kazan, Russia; ostefzueva@mail.ru

**Keywords:** confined water, reverse micelles, dielectric relaxation, “defect” models, hydrogen bonds

## Abstract

Relaxation behavior of water confined in reverse micelles under temperature-induced micelle clustering is undertaken using broadband dielectric spectroscopy in frequency range 1 Hz–20 GHz. All microemulsion systems with sufficiently noticeable micelle water pool (water/surfactant molar ratio *W* > 10) depict three relaxation processes, in low, high and microwave frequencies, anchoring with relaxation of shell (bound) water, orientation of surfactant anions at water-surfactant interface and relaxation of bulk water confined in reverse micelles. The analysis of dielectric relaxation processes in AOT-based w/o microemulsions under temperature induced clustering of reverse micelles were made according to structural information obtained in NMR and conductometry experiments. The “wait and switch” relaxation mechanism was applied for the explanation of results for water in the bound and bulk states under spatial limitation in reverse micelles. It was shown that surfactant layer predominantly influences the bound water. The properties of water close to AOT interface are determined by strong interactions between water and ionic AOT molecules, which perturb water H-bonding network. The decrease in micelle size causes a weakening of hydrogen bonds, deformation of its steric network and reduction in co-operative relaxation effects.

## 1. Introduction

Since the first experimental measurements of the dielectric constant of water, almost every new experimental study brings additional interest to its dynamic structure and properties. We have undertaken the attempt to look at the non-Debye relaxation behavior of water confined in reverse micelles using the “wait and switch” relaxation model [1]. Reverse micelles, the constituent of water-in-oil (w/o) microemulsions, permit one to alter the level of water confinement, varying the water-to-surfactant ratio and temperature.

The unique properties of liquid water are highly determined by its natural gift to build three-dimensional lace of hydrogen bonds. The unique properties of water such as hydrogen bonding of its molecules determines the bulk structure of water containing countless molecular clusters and structural assemblies with different size and time scales. The hydrogen bond network between water molecules results in its tetrahedral ordering [2,3]. During thermodynamic oscillations hydrogen bonds promote such irregularities in water tetrahedral ordering as bifurcated, trifurcated H-bonds and other, more complicated configurations [4], which can be defined as the orientation defects [5] (Figure 1A). In addition to orientation defects in water, there are also ionic defects present which are formed due to the migration of H_3_O^+^ and H^−^ pairs (Figure 1B). It is obvious that the probability of these processes is dependent on the properties of the water-containing system, the presence of solutes and different boundaries.

The direct way to watch water dynamics is to study its dielectric response or dielectric relaxation [6]. The main water dielectric relaxation process obeys the Debye-like dispersion up to a hundred gigahertz [7], which reflects cooperative reorientation of water dipoles [1,5,8,9]. One of the first models of dielectric relaxation in water was proposed by P. Debye [10]. Now this model with water molecules performing the small-angle Brownian rotational motion is considered as erroneous [11] because of the hydrogen bonds army which limits the free rotation of water molecules. As the alternative for water molecules to change sequentially the orientation of the dipole moment is the stepwise jumps with disruption and “re-switching” of hydrogen bonds. To describe the abrupt reorientation of water molecules, the most popular now is the “wait and switch” relaxation model [1], according to which the reorientation of water molecule occurs only at large angles when it encounters a defect in the hydrogen bond network. Based on this polarization mechanism, a new theoretical model describing the dielectric relaxation of bulk liquid water was proposed [5]. It is necessary to note that various “defect” models of dielectric relaxation have appeared much earlier [12,13] than the “wait and switch” one and were successfully applied to explain dielectric experiments [14].

In the present work we studied dielectric relaxation of water confined in the system of reverse micelles [15]. Geometrical confinement significantly influences the dynamics of water in small cavities, as a result of its interactions with interfaces [16]. However, it is not completely understood how the confinement of different nature, structure and size could affect collective modes of water dielectric relaxation [17]. Thus, we were interested to look at the water dielectric behavior in the system where it is possible to alter the confinement size and structure via alterations of system composition and temperature. For this purpose, we have selected the surfactant microemulsions, which are smart systems to construct various supramolecular architectures [18,19,20,21]. The w/o microemulsions are the dispersions of reverse surfactant micelles each carrying a water pool in its center [15,22]. The size of the water pool is proportional to the water/surfactant molar ratio, so it is rather easy to change the micelle size [23] and the ratio between the bulk and the interfacial water species [24]. Another interesting phenomenon in the w/o microemulsions is the temperature-induced clustering of reverse micelles which is clearly highlighted in the electric percolation phenomenon [25], when reverse micelles start to associate at the definite temperature, merging the water cores of adjacent micelles and bringing new conditions for water being and ion transport [26]. So, the main aim of the present research was to study dielectric relaxation of confined water in the bulk and interfacial states and look at water relaxation properties in the reverse micelle confinement under their temperature-induced clustering.

## 2. Results

### 2.1. Structural Features of AOT-Based Water-in-Oil Microemulsions

We have chosen the AOT-based w/o microemulsions to study the dielectric relaxation of water, confined inside surfactant reverse micelles, composed of nanosized water droplets surrounded by a surfactant monolayer and dispersed in continuous organic phase (oil) [27,28]. The AOT-based reverse micelles permit one to alter micellar size and the ratio between free and bound water. Among various surfactants [29,30], the sodium bis(2-ethylhexyl) sulfosuccinate (AOT) takes up its specific place due to original chemical structure and properties [31]. This short, double-tail anionic surfactant with trapezoid-like shape forms rather strong monolayers, which stabilize water–oil interface in a broad range of water-to-oil ratio [32]. Temperature plays a significant role in microemulsion stability. It influences the solubility of surfactant hydrophobic and head fragments in water and oil phases, subsequently influencing the phase stability of microemulsion. AOT is known as the reagent which stabilizes microemulsion in a very broad range of temperature and water-to-surfactant ratio [33]. The choice of decane as the dispersive medium was provided by its moderate volatility at increasing temperatures and preservation of physical–chemical stability of microemulsion at varying temperatures.

The radius of the water core of AOT reverse micelle depends on the degree of surfactant hydration [34], which is defined as the dimensionless water/surfactant molar ratio *W* = [H_2_O]/[surfactant]. The radius of micellar aggregates $R_{M}$ (nm) is estimated using a geometric model, based on the balance between the volume of water pool and the surface of surfactant shell [35,36]:(1)$$R_{M}\;=\;0.15\;W\;+\;L_{\mathrm{AOT}},$$
where $L_{\mathrm{AOT}}$ is the length of AOT molecule (~1.4–1.5 nm). The countless attempts to determine the size of AOT micelles were very popular in the scientific world for many decades using DLS [37,38], small-angle X-ray scattering (SAXS) [39,40], fluorescence correlation spectroscopy [41] and ^1^H NMR-PFG [23,42]. One can find in the literature AOT-based micelle radii in the range 2–20 nm, and even up to 42 nm [43] as a function or *W* and micelle volume fraction, but the origins of these differences are still unclear. Nevertheless, this is not the goal of the present work to discuss possible differences in the precise estimation of the micelle size from different experimental methods, it is quite enough for us to realize that the radius of the water pool of AOT-based reverse micelle is roughly proportional to W, varying in our study by about one order of magnitude or three orders in the volume, influencing the water properties significantly [44].

Our experimental work on the determination of the micelle size in AOT w/o microemulsions entirely confirms this situation. Figure 2 depicts the results of the micelle size obtained by two independent methods, DLS and ^1^H NMR-PFG.

One can see that for small and moderate micelles (*W* = 4–15), there is very good accordance in the hydrodynamic radii of reverse micelles obtained by different techniques. However, for larger W there is considerable deviation from the initial linear course of obtained dependence. Another result, as shown in Figure 2, is that Equation (1) gives another slope of dependence, i.e., the coefficient at *W* in the theoretical equation differs about 3.3 times from the experimental slope. The theoretical values of *R_M_* from Equation (1) are lesser (2.5–10 nm), since the geometrical model does not take into account thermal motion which can increase the real size of micelles.

We anticipate that the increased experimental size of AOT reverse micelles in comparison with the theoretical one can be the result of micelle clustering with the increase in temperature or micelle concentration. The physical characteristics that mostly clearly reflect the specific behavior of w/o microemulsions during microdroplets clustering are the electrical conductivity and diffusion mobility of its components, leading to the percolation phenomenon. When considering the phenomenon of electrical percolation in w/o microemulsions, one should remember that the conductivity of oil phase is manifold lower than that of the dispersed phase. The free charges in the dispersed phase are the ionized surfactant molecules and their counterions. In general, the statistical ensemble is electrically neutral, but due to thermal fluctuations, microdroplets can carry a charge. The electrical conductivity of such systems is determined by diffusion mobility of micelles that have an uncompensated electrical charge. The diffusion of micelles in a viscous medium obeys Stokes’ law, and the electrical conductivity σ is depicted as [38](2)$$\sigma =\frac{\varepsilon_{0}\varepsilon k_{B}T}{2\pi \eta }\frac{\phi_{M}}{r^{3}},$$
where $\varepsilon_{0}$ = 8.85 × 10^−12^ F/m is the dielectric constant, ε is the dielectric permittivity of organic medium, $k_{B}$ is the Boltzmann constant, *T* is the absolute temperature, η is the viscosity of the organic medium and $\phi_{M}$ is the volume fraction of the dispersed phase. During thermal motion, reverse micelles collide with each other and exchange by surfactant ions and water pool content containing counterions. During the exchange of matter between adjacent particles, the charge carriers are transferred in the volume of microemulsion. With increasing temperature, there is the growth of mobility of micelles and the probability of their contacts with the interlocking of adjacent micellar aggregates by surfactant hydrocarbon radicals and the exchange of micelle content through the coalescence mechanism. At a certain temperature, the clusters of reverse micelles are formed, which creates a path for charge transfer during formation of short-living channels between neighboring micelles, causing a sharp increase in the electrical conductivity of the microemulsion (Figure 3). The formation of conducting paths in an insulating material is called the electrical percolation phenomenon [45]. The process of cluster formation by reverse micelles has a dynamic nature. They are formed and destroyed in the dispersion bulk under the influence of thermal motion. Above a certain temperature, named the percolation threshold *T*_P_ (determined as the temperature of derivation maximum in Figure 3B), in the volume of microemulsion there is always at least one “infinite” cluster, which ensures the constancy of high electrical conductivity of the system [46]. Comparing the results presented in Figure 2 and Figure 3 it is obvious that at 30 °C most of studied systems are far from the state of separated single micelles. Thus, it is not surprising that diffusion methods are giving size values enlarged in comparison with theoretical predictions. Obviously, it is a positive result for the main goal of our work—to detect correlation between the geometry of water confinement and dielectric properties upon clustering, because we have varying water distribution in the system with increase in temperature.

Another characteristic reflecting the clustering of reverse micelles and the changes in the water state is the diffusion mobility of microemulsion components, which, like electrical conductivity, reflects the transfer of matter in the volume. Analyzing the data on the self-diffusion coefficients of water and AOT obtained in proton NMR experiment (Figure 4), one has to take in mind that the molecules of different compounds may take part in different diffusion motions due to the fast exchange of molecules between different structural states [23,47,48]. A specific feature of ^1^H NMR experiment with surfactant systems is a relatively long measurement time up to a hundred milliseconds. The ^1^H NMR experimental time scale is long enough for surfactant molecules and solubilized water to undergo multiple transitions from the monomeric to micellar state and between micelles, resulting in the weight-averaged measured parameters according to the two states model [47,49]. For the studied system, the surfactant and water molecules can diffuse together with micelles having slow self-diffusion coefficient *D*_M_ of micelle and in parallel there can be a fast diffusion of individual molecules across the clusters during material exchange between micelles. Here, it will be useful to note that to detect this fast motion in proton NMR experiment, the diffusion path (the size of cluster) has to be lengthy enough. This is seen quite exactly in Figure 4A in the dependence of water and surfactant self-diffusion coefficients on the size of micelles. Below *W* = 15, for small and moderate reverse micelles, one can see almost the linear dependence of surfactant self-diffusion, which reflects the consecutive growth of micelle size as shown in Figure 3. Under the further increase in W, one can see the acceleration of surfactant diffusion, evidently due to faster self-diffusion of individual surfactant molecules within micelle clusters. At the same time, the diffusive mobility of water increases even more strongly due to the small size of water molecules and their high mobility along micelle clusters. These observations show that at W > 15, the size of clusters is high enough to form some specific phase for the diffusion of water and surfactant. In addition, one can propose the formation of short-living water channels between adjusting molecules (see below). In any case, with an increase in the radius of reverse micelle and with the temperature-induced micelle clustering, we see (Figure 4B) the transition of water state towards the bulk one (chemical shift ~4.8 ppm at room temperatures) [50,51] due to the increase in the volume of micelle water pools and the formation of water channels. We can conclude that according to chemical shift data, the maximal difference in water properties in reverse micelles from the bulk ones is determined for the minimal size of water core (*W* = 4) with a sharp alteration towards free bulk water at the moderate *W* and the slow dependence with the following increase in micelle size and their clustering.

### 2.2. Dielectric Relaxation

The w/o microemulsions including the AOT-based systems are very popular objects for dielectric studies for many years [52,53,54,55,56,57]. It seems that, to date, everything is clear for the AOT-based w/o microemulsions from positions of dielectric relaxation. Indeed, it appears that this system is rather clear and explainable for dielectric study. At temperature slightly lower than the room one the system consists of nanosized droplets dispersed in a continuous oil phase, organized predominantly from normal or isomeric alkanes C_6_–C_10_. The droplets (reverse micelles) are organized from the water pool surrounded by a monolayer of AOT surfactant molecules (Figure 5). This water is divided into two fractions, the core, where the properties of water are approaching the bulk one, and the shell, where water is under the strong interaction with surfactant interface. Molecules of AOT dissociate into anions SO_3_^−^ in their charged head groups and Na^+^ counterions. As a result, in such a system there are three main sources of polarization, which give corresponding relaxation processes. They are the bulk (core) and the bound (shell) water, and also the charge rearrangement relative to the water–surfactant interface, although the last one may be of a different origin proving oneself in different frequency ranges.

Unfortunately, even in spite of this apparent simplicity of microemulsion systems, they are not simple for dielectric studies. First of all, it follows from the nonselective nature of electric (dipole) polarization processes. Second, the dielectric relaxation phenomena are too broadband in their nature with complicated reciprocal overlapping. Third, the systems based on the ionic surfactants are overcrowded by different electric charges, which “close” relaxation processes by straight-through conductivity and respond actively to their interaction with the interface. Fourthly, there is a dramatic structural rearrangement of microemulsion under the influence of components ratio and temperature. Thus, as a whole, these are the reasons that many of the obtained experimental results for microemulsion systems are difficult for comparison. To overcome these complications, we tried to combine in the present research the sweeping of two parameters. They are the size of reverse micelles, which alter the ratio between free and bound water, and the temperature. Another problem of adequate description of relaxation processes in the AOT-based w/o microemulsions arises in many cases due to a partial covering of the frequency range, which is required to capture all main dielectric processes. In this sense, the present study aims to rise up to microwave frequencies to have relaxation of bulk water as a reference process.

The typical overview of obtained relaxation processes is presented in Figure 6. All systems with *W* > 10 depict three relaxation processes, which we labeled as low-frequency (LF), high-frequency (HF) and microwave (MW) ones. The spectra were approximated using the superposition of three Cole–Cole terms, conductivity contribution and the Jonscher function, which describes dielectric behavior at a long-time limit [57,58,59,60]:(3)$$\varepsilon^{*}\left(\omega \right)=\varepsilon_{\infty }+\frac{\Delta \varepsilon_{\mathrm{LF}}}{1+{\left(i{\omega \tau }_{\mathrm{LF}}\right)}^{\alpha_{\mathrm{LF}}}}+\frac{\Delta \varepsilon_{\mathrm{HF}}}{1+{\left(i{\omega \tau }_{\mathrm{HF}}\right)}^{\alpha_{\mathrm{HF}}}}+\frac{\Delta \varepsilon_{\mathrm{MW}}}{1+i{\omega \tau }_{\mathrm{MW}}}+\frac{\sigma_{\mathrm{DC}}}{i{\omega \varepsilon }_{0}}+\frac{u_{\mathrm{EP}}}{\omega^{n}},$$
where $\varepsilon_{\infty }$ is the high frequency limit of dielectric permittivity, subscripts LF, HF and MW denotes the low-frequency, high-frequency and microwave dielectric relaxation processes, Δε, τ and α are the dielectric strength (amplitude of dispersion), relaxation time and relaxation time distribution (0 < α < 1), $\sigma_{\mathrm{DC}}$ is the through conductivity, $u_{\mathrm{EP}}$ and *n* are the amplitude and exponent of Jonscher function, respectively.

It is easier to start the identification of detected relaxations starting from the most high-frequency process (MW) centered about dozens of GHz. Its frequency range is typical for relaxation of bulk water [61]. The next significant evidence is the value of activation energy of MW process obtained from corresponding Arrhenius plot 16 kJ·Mol^−1^ (Figure 7, which is very close to the known value of activation energy for the pure water dielectric relaxation of 15.9 kJ·Mol^−1^ [62,63]. An additional evidence of the assignment of MW process to the relaxation of bulk water (water core in Figure 5) is its linear dependence within 5–55 °C temperature range, because the bulk water, if present, has no motives to alter its dynamical structure and mobility upon possible structural transformations of reverse micelles (see below).

The joint view on LF and MW relaxation processes, presented in Figure 7 and Figure 8, allows us conclude that they relate to two subensembles of water entrapped inside reverse micelles: the bulk-like water core and a hydration shell near the ionic surfactant head groups, schematically shown in Figure 5. These two distinct water species display different polarization kinetics, reflected in relaxation time τ and dielectric strength Δε. The relaxation rates of these water species are mutually independent variables, reflecting different structural and interaction arrangement of water dipoles. Nevertheless, the dielectric strengths of two processes are interdependent; they are proportional to relative populations of core and shell water species, which correlate with the volume-to-surface ratio of micelle water pool.

Under the variations in micelle size, we have the disproportional alteration of two water populations, which is the result of interplay between the area of spherical surface *S*_SHELL_ and ball volume *V*_CORE_:*S*_SHELL_ = 4π(*R_W_* − δ/2)^2^;           *V*_CORE_ = (4/3)π(*R_W_* − δ)^3^,(4)
where *R_W_* is the radius of water pool of reverse micelle and δ is the width of the hydration water shell. In the first approximation the ratio of the core and shell species is proportional to *R_W_* or *W*. At that, it should be kept in mind that this proportion works better for big *W* values or large micelles. As a whole, one can detect such tendency in Figure 8LF,MW), since dielectric strength Δε is proportional to the number of dipole moments, i.e., to water population in two species. One can see the coincidence of $\Delta \varepsilon_{\mathrm{LF}}$ decrease and $\Delta \varepsilon_{\mathrm{MW}}$ increase for varying *W* from 15 to 35, known also from other experimental techniques [64]. This is the convincing argument, together with linear Arrhenius dependence of $\tau_{\mathrm{LF}}$ (Figure 7LF, that the LF process is determined by the relaxation of shell (bound) water. An additional reason is the absence of core (bulk) water in small micelles (*W* = 10).

The additional interest in the study of reverse micelles is their temperature-induced clustering clearly detected in our conductivity (Figure 3) and HF relaxation (Figure 7HF and Figure 8HF data as their sharp breaks at percolation threshold. Under existent knowledge, the HF relaxation process is determined by polarization of surfactant [54,55]. However, since this study is mainly devoted to water properties, we will not focus special attention on this process but only the determined HF relaxation process, centered around 100 MHz, which arises due to the correlation displacement of surfactant polar head groups surrounding each water pool of reverse micelles, which differs from the interfacial Maxwell–Wagner effect usually observe at much lower frequencies [65]. The determined dielectric dispersion is the result of the AOT head group motion with respect to the hydrophobic part of the surfactant molecule at the water–oil interface. This motion appears in the fluctuating electric dipole moment of surfactant interface, which gives rise to detected dielectric relaxation. One can see that both the dielectric strength Δε and relaxation time τ are very sensitive to the structural rearrangement of surfactant monolayer upon reverse micelles clustering (Figure 7HF and Figure 8HF) with two well-known effects, concerning the maximum of ε(ω) and τ close to the percolation threshold. Below percolation threshold one can see the sharp increase in Δε (Figure 8HF), attributed to increase in correlated orientation of AOT anions at the water-surfactant interface [66]. The increase in temperature leads to an increase in relaxation time below the percolation threshold (Figure 7HF). Very likely, it is a result of permanent alteration of surfactant microenvironment and decrease in surfactant mobility [54]. Above the percolation transition, we see the inverse picture with a sharp decrease in both the Δε and τ of HF process, typical for the dielectric relaxation process in structurally conservative systems [67].

Let us schematically overview the key structural rearrangement stages of the AOT-based water/oil (decane) interface (Figure 9). Under initial temperature growth the reverse micelles come into the short-living contacts (*T*_2_), then forming increasingly big clusters with long water channels providing the percolation phenomenon (*T*_3_). At the further increase in temperature, the number of such infinite clusters is growing with the formation of reverse tubes (*T*_4_), following phase layering at *T*_5_. As one can see in Figure 3, the percolation transition is rather symmetric in temperature. Thus, we have roughly equal temperature intervals below and after the percolation threshold. During the first interval, there are permanent alterations in the micelle structure including the changes in average interface bending. In other words, there is the ongoing realignment of polarization and relaxation conditions of surfactant head groups with the rise in temperature till the percolation transition, which become apparent as the “difficult-to-explain” rise in Δε and τ with temperature growth. At the second stage, from *T*_3_ up to phase layering at *T*_5_, the system consists mainly of long chains of permeable reverse micelles or even from some kind or reverse cylindrical micelles or tubes [68,69]. In this temperature range, one has only the elongation of pseudo tubes, which preserves surfactant structural conditions invariable and the theoretically “correct” decrease in Δε and τ under the rise in temperature.

Proceeding to the analysis of water behavior, we have to underline, that its relaxation course both for core and shell water fractions does not notice micelle clustering and possible structural transformation of micelles. The core water is rather close to the free bulk one in the values of relaxation time and activation energy, even taking into account the observed “blue shift” or faster reorientation of water apparent dipole moment due to ion-dipole interaction of water with Na^+^ counterions [61].

The dynamic behavior of shell water is very sensitive to the size of reverse micelles. The relaxation time of this water fraction strongly increases with the growth of water pool size (Figure 10B). Temperature dependences of LF relaxation time are linear in Arrhenius coordinates, i.e., the dipole mobility of shell water obeys the activation mechanism. For *W* = 10 we see the shortest times and the highest activation energy of 91 kJ/mol. By the increase in water pool size one can detect the increase in relaxation time with decrease in activation energy up to of 44 kJ/mol at *W* = 35. Evidently, this is a result of increase in the surfactant interface bending accompanied by rearrangement of surfactant packing and interaction with bound water under the progressive divergence of tetrahedral ordering of bound water molecules and disruption of hydrogen bond network near the H_2_O/AOT interface [64].

In the beginning, we gave a brief overview of modern tendencies in the dielectric relaxation of water in the liquid state. The most adequate model, which satisfactorily describes the relaxation of water is the “wait and switch” model [5,6,11]. It is a consequence of a tight network of hydrogen bonds, which restricts the rotational diffusion of water molecules. Therefore, the reorientation of dipole moments cannot occur freely as in the gaseous state, but rather consists of abrupt jumping of water dipole moments as a result of hydrogen bonds disruption (Figure 1A). The rupture of a hydrogen bond leads to orientational defect formation. This process requires definite time to restore the broken hydrogen bond and a wait for the next defect event, being the specific characteristics of certain system. The defect migration has an intermittent character being analogous to the Bjerrum’s L-D orientation defects [1,70]. In water the orientation defects are formed in pairs (defect–antidefect) and are destroyed upon their meetings. In addition to orientation defects, there are also the ionic defects OH^−^/H_3_O^+^ appearing via proton hopping in water [71]. Due to the ionic defects, water molecules also rearrange their dipole moments according to proton jumps along hydrogen bonds (Figure 1B). In this case, if a proton is accepted from one side of the water molecule, it is given back from the other side.

The “wait and switch” relaxation mechanism can be applied for explanation of obtained results for water in the bound and bulk states under spatial limitation in reverse micelles. In small micelles (*W* ~ 10), where the fraction of core water does not exceed 10–15% of water pool [64], the strong correlation of dipole moments leads to the strengthening of hydrogen bonds and the high values of activation energy (Figure 10B). With increase in the water pool volume, the correlations of water dipole moments decrease with decrease in spatial limitations and mobility of water dipole moment slows down.

The degree of dipole moment correlation determines the dielectric strength of process Δε, which are shown in Figure 10A. The Δε values are maximal for *W* = 10 ÷ 15, decreasing to *W* = 25 ÷ 35. The shape of the curves Δε (*T*) is characteristic for melting of strongly associated liquids, such as water and alcohols [72]. With increase in temperature, the mobility of dipole moments increases and reaches maximum, then at percolation threshold the neighboring micelles start the exchange by water molecules up to formation of lengthy channels with a sufficiently long lifetime.

By analyzing the obtained results and comparing them with the known regularities of water hydrogen bonding in reverse micelles we can make the following confirmations of our conclusions. For example, early it was shown that water properties in the proximity of the AOT surface differ from the properties of bulk water [64]. The AOT layer influences the water properties only locally, over a distance shorter than 0.4 nm [73] and thus the surface affects only one or two molecular layers. This confirms our argumentation on the numeral ratio between the core and shell water (Equation (4)) and their contributions to LF and MW relaxation processes. Also, we proposed that water properties at the AOT interface are determined by strong interactions between water and ionic AOT molecules, which was known from MD simulations [74,75] and NMR studies [69]. Moreover, it was reported [76] that these interactions perturb the H-bonding network. The decrease in micelle size causes a restructuring of the hydrogen bond network [77,78] with decrease and weakening of hydrogen bonds when they are confined in smaller droplets [79,80], causing the reduction in cooperative effects through the deformation of hydrogen bond network.

## 3. Materials and Methods

### 3.1. Chemicals

Water-in-oil (w/o) microemulsions were prepared using the anionic surfactant sodium bis(2-ethylhexyl) sulfosuccinate (AOT), produced by TCL (Belgium), product number S0139. Surfactant in concentration 0.35 M was dissolved in decane of 99% purity (Acros Organics, Moscow, Russia). Then, the appropriate amounts of deionized Milli-Q water (purified by “Arium mini” ultrapure water system, Sartorius, Gottingen, Germany) were added to keep dimensionless molar ratio equal to *W* = [water]/[AOT] = 4, 6, 10, 12, 15, 20, 25, 30 and 35, varying the size of AOT-based reverse micelles [51].

### 3.2. Dielectric Measurements

The dielectric experiments in 10 frequency decades from 1 Hz to 20 GHz were carried out in three stages. At the first stage, the dielectric spectra of microemulsions were registered using the Alpha Frequency Response Analyzer as a part of the Novocontrol BDS-80 measuring complex. The results were recorded with the built-in licensed WinDeta software package (Version 2.0). The frequency range of measurements was 1 Hz–10 MHz. A plane-parallel capacitor with an electrode diameter of 12 mm was used as a measuring cell, the distance between electrodes was set by a fluoroplastic washer with a thickness of 0.5 mm. Calibration of cell measurement was carried out at 20 °C using air and benzene.

At the second stage, the measurements were carried out using the E4991A radio frequency analyzer included in the Novocontrol BDS-80 measuring complex, in the frequency range of 1 MHz–1 GHz. The samples were placed in a similar measuring cell.

At the third stage, the dielectric spectra were measured on the PNA-X Agilent N5247A Network Analyzer in a frequency range of 1 GHz–20 GHz. The results were recorded using the built-in licensed Agilent 85070 software package. A coaxial performance probe with a diameter of 10 mm was used as the measuring cell. It was calibrated using the deionized Milli-Q water at a temperature of 20 °C.

The temperature range of measurements in the 5–50 °C range were made in steps of 5 °C. Temperature control was carried out using the Quatro system during the two first stages of measurements and LOIP LT 900 thermostabilizer during the third stage. The spectra obtained at three measurement stages were combined to obtain the broadband spectra in the frequency range of 1 Hz to 20 GHz. The dielectric relaxation parameters were calculated using the Datama software package (version 2.0) [81].

### 3.3. H NMR Experiments

NMR spectroscopy is often used to study the dynamical structure of surfactant systems via spectral analysis and self-diffusion measurements [47,49,82]. Here we used the pulsed-field gradient (PFG) proton NMR to analyze chemical shifts and self-diffusion coefficients of microemulsion components to estimate the reverse micelles size in microemulsions [23]. The proton NMR experiments were performed on a Bruker AVANCE III NMR spectrometer operating at 600.13 MHz, equipped with an inverse triple resonance high-resolution probe (TXI, 5 mm), z-gradient and the BCU05 temperature control unit. The self-diffusion coefficients were measured using the modified Stejskal–Tanner pulse sequence “selgpse”. In the used sequence, the RF 180°-pulse had a Gaussian shape with a duration of 80 ms. The diffusion time was 200 ms. The duration of the sinusoidal gradient pulse was 7 ms, the gradient amplitude varied from 0 to 0.55 T/m. The self-diffusion coefficient was calibrated using pure water with a self-diffusion coefficient of 2.6 10^−9^ m^2^/s at 30 °C. In all diffusion experiments, a mono-exponential decay of the resonance line intensity was observed within three orders of magnitude with a correlation coefficient ≥ 0.995. The external coaxial insert with acetone-d6 (Aldrich, Saint Louis, MI, USA, 99.9%) was used to stabilize the magnetic field. All self-diffusion measurements were made at 30 °C. Data processing and analysis were performed using the Bruker Topspin 3.6.1 software.

### 3.4. Dynamic Light Scattering

The dynamic light scattering (DLS) of reverse micelles were studied with the Malvern Instrument Zetasizer Nano. The He–Ne laser with 4 mW power and 633 nm wavelength was used as a source of light irradiation. Measurements were performed at 173° scattering angle. The registered autocorrelation functions were analyzed by the Malvern DTS software (version v1.41). The micelle effective hydrodynamic radius (*R*_H_) was calculated according to the Stokes−Einstein equation by the built-in software package. The temperature of measuring cell was controlled at 30 °C. The DLS experiments involved no less than five measurements in ten runs. The samples were cleaned from impurities using the 0.2 μm membrane filters.

### 3.5. Electric Conductivity of Microemulsions

The low-frequency electrical conductivity was measured in the temperature range 10–50 °C using a Radelkis OK102/1 conductometer (Budapest, Hungary) at 100 Hz and 3 kHz.

## 4. Conclusions

We have undertaken the attempt to look at the non-Debye relaxation behavior of water confined in reverse micelles. For this purpose, we applied the broadband dielectric spectroscopy and obtained relaxation data in the frequency range from 1 Hz to 20 GHz. All systems with sufficiently noticeable water pool (*W* > 10) depict three relaxation processes, in low, high and microwave frequencies, anchoring with the relaxation of shell (bound) water, orientation of surfactant anions at the water-surfactant interface and relaxation of bulk water confined in reverse micelles, correspondingly. The spectra were analyzed using the superposition of three Cole–Cole terms, conductivity contribution and the Jonscher function, describing dielectric behavior at low frequencies. The analysis of dielectric relaxation processes in the AOT-based w/o microemulsions under temperature induced clustering of reverse micelles were made with the help of structural information obtained by NMR and conductometry experiments.

We have shown that the “wait and switch” relaxation mechanism can be applied for the explanation of dielectric relaxation of water in the bound and bulk states under spatial limitation in reverse micelles. Analyzing the obtained results, we made the following conclusions. The AOT layer influences mostly the properties of bound water. Water properties at the AOT interface are determined by strong interactions between water and ionic AOT molecules, which perturb the H-bonding network. The decrease in micelle size causes a weakening of hydrogen bonds, deformation of its steric network and reduction in cooperative relaxation effects.

## Figures and Tables

**Figure 1 ijms-26-07152-f001:**
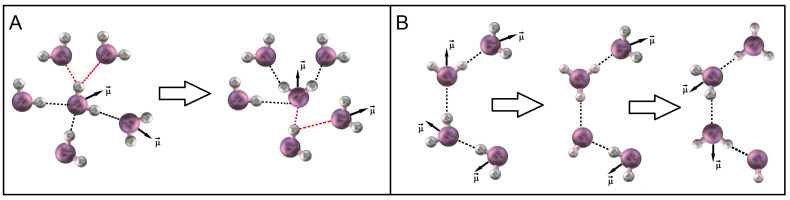
Bifurcated orientation defects (highlighted in red) in water tetrahedral ordering (**A**) and ionic defects formed by migration of H_3_O^+^ and HO^−^ pairs (**B**).

**Figure 2 ijms-26-07152-f002:**
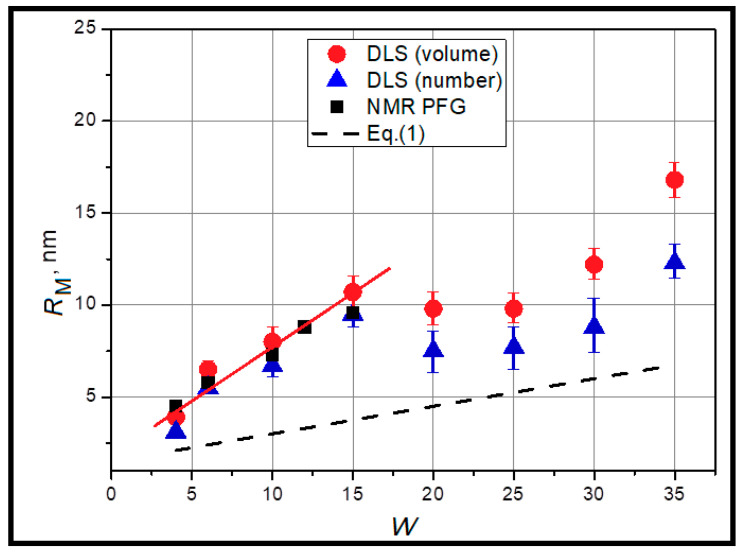
Experimental (DLS and ^1^H NMR PFG) and theoretical values (Equation (1)) of micelle size at 30 °C. Red straight line is shown as a guide for eyes. Size distribution by volume (red) and number (blue). Polydispersity of DLS results is shown as data spread relative to the average of 5 measurements.

**Figure 3 ijms-26-07152-f003:**
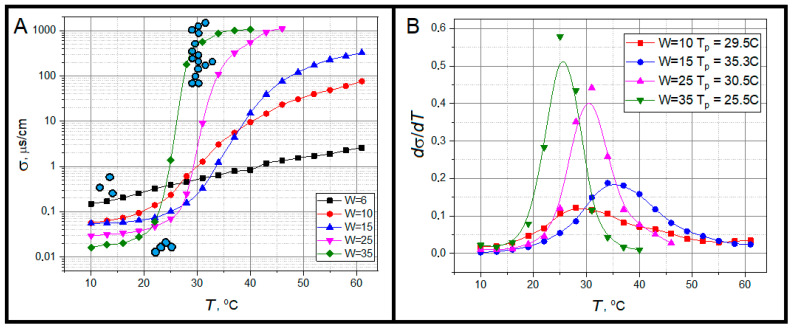
Temperature dependences of electric conductivity (**A**) and its first derivative (**B**) for AOT-based o/w microemulsions. In (**A**) micelle (blue circles with black bordering) clustering is shown schematically.

**Figure 4 ijms-26-07152-f004:**
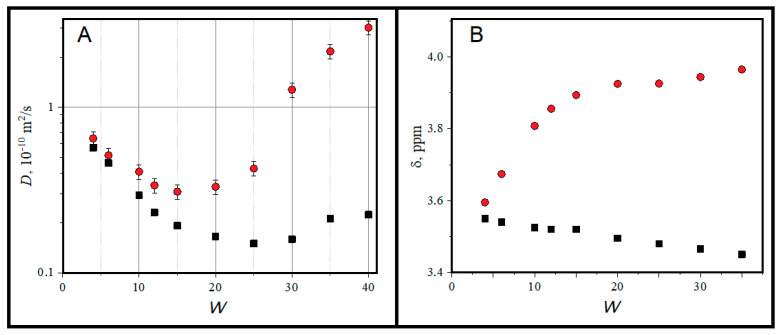
Self-diffusion coefficients (**A**) and chemical shift (**B**) for water (red circles) and AOT (black rectangles) as a function of W value.

**Figure 5 ijms-26-07152-f005:**
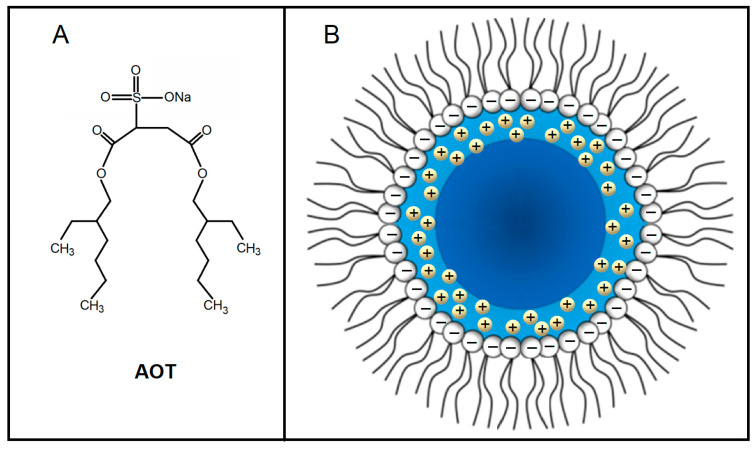
Chemical structure of AOT (**A**) and schematic structural scheme of AOT-based reverse micelle (**B**). Core (dark blue) and shell (light blue) regions of water pool of AOT-based reverse micelle with dissociated Na^+^ counterions.

**Figure 6 ijms-26-07152-f006:**
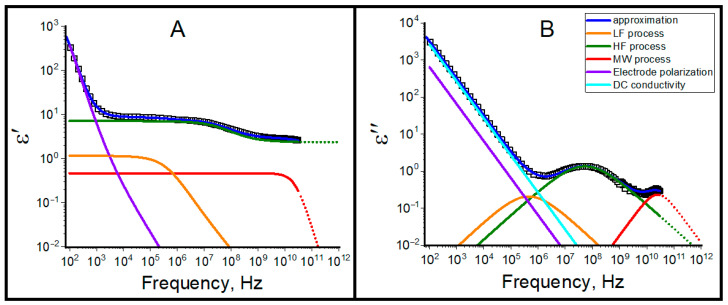
Experimental data (squares) and its fitting by Equation (3) for microemulsion *W* = 15 at 20 °C: (**A**)—dielectric permittivity; (**B**)—dielectric losses.

**Figure 7 ijms-26-07152-f007:**
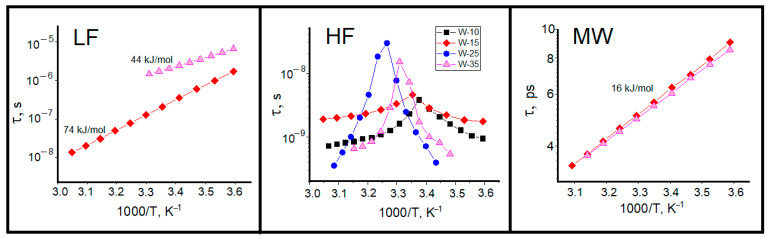
Arrhenius plots of dielectric relaxation times for (**LF**), (**HF**) and (**MW**) processes in AOT-based w/o microemulsions.

**Figure 8 ijms-26-07152-f008:**
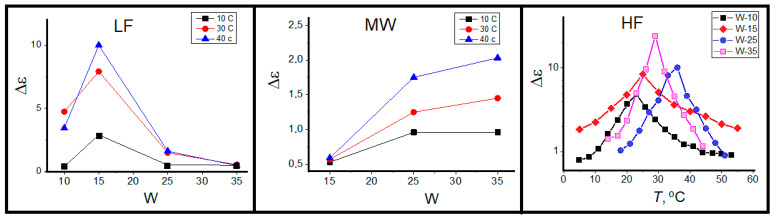
Dielectric strength Δε of (**LF**), (**MW**) and (**HF**) relaxation processes as a function of *W* and temperature.

**Figure 9 ijms-26-07152-f009:**
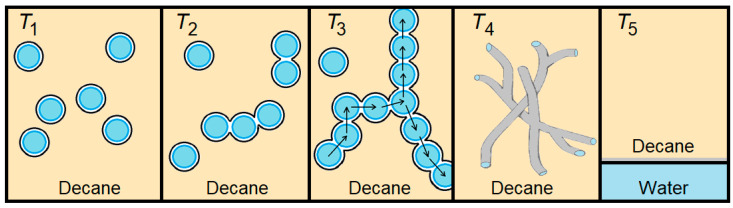
Scheme of temperature-induced structural transformations in w/o microemulsion: intact system (*T*_1_), micelles clustering (*T*_2_), percolation (*T*_3_), tubular structures (*T*_4_) and phase layering (*T*_5_). Blue circles with black bordering represent schematically reverse micelles. Arrows show the transfer of substance over micellar clusters.

**Figure 10 ijms-26-07152-f010:**
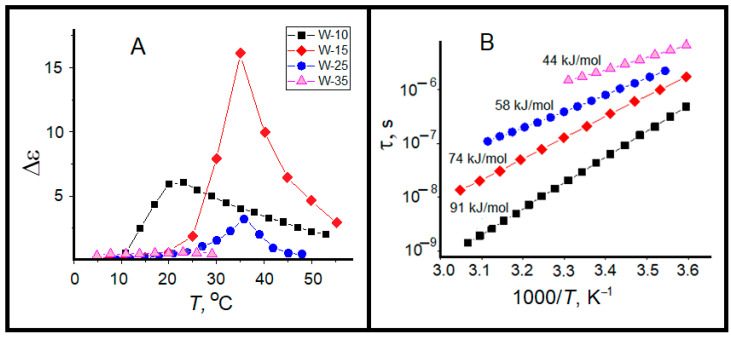
Parameters of shell water relaxation (LF): (**A**)—dielectric strength Δε, (**B**)—relaxation time τ.

## Data Availability

The data in this study are available on reasonable request from the corresponding author.

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
