# Peer review of "The Relaxation Behavior of Water Confined in AOT-Based Reverse Micelles Under Temperature-Induced Clustering"

_ijms, 2025, doi:10.3390/ijms26157152_

Round 1
Reviewer 1 Report
Comments and Suggestions for Authors
The work by Lunev et al explores the relaxation behavior in reverse (water-in-oil) emulsions stabilized by AOT surfactant. The main result shown from the extensive set of experiments is that the “wait and switch” relaxation mechanism can be applied for explanation of dielectric relaxation of water in the bound and bulk states under spatial limitation. In turn, the properties of bound water were found to be mostly influenced by surfactant heads which perturb the H-bonding network. While the whole text is consistent and clearly written, its predictive power should be enhanced. Namely, regarding the selected type of surfactant, a natural question arises: how would its molecular architecture influence the bonding strength of the interface? For insrtance, what would happen if one chooses the other type of surfactant, say, the one with a single tail, larger head or even two heads (the case of gemini surfactants)? Furthermore, what would happen if one chooses the organic solvent with higher compatibility with water (say, by creating emulsions using 1-octanol or toluene)? I believe that there should be a part of the text where such issues are discussed. Addittionally, I suggest to color the schematic aggregates in Figure 3 to evenly differ them from the other symbols on the graph. Again, regardring the rest of the text, everything else is fine and suitable for publishing in the current journal.
Author Response
Dear Reviewer,
Thank you for valuable comments and recommendations. We have added the following fragment (colored in blue in pdf format of revised article) to Section 2.1. Structural Features of AOT-based Water-in-Oil Microemulsions, giving answers to your comments with appropriate references:
“Among immense army of surfactants [], the sodium bis(2-ethylhexyl) sulfosuccinate (AOT) takes up its specific place due to its original chemical structure and properties []. This short, double-tail anionic surfactant with trapezoid-like shape forms rather strong monolayers, which stabilize water-oil interface in a broad range of water-to-oil variations []. Temperature plays a significant role in the microemulsion stability. It influences the solubility of hydrophobic and head fragments of surfactant in water and oil phases, subsequently influencing phase stability of microemulsion. AOT is known as the substance which can keep microemulsion state in a very broad range of temperature changes and water-to surfactant ratio []. The choice of decane as the dispersive medium was provided by its moderate volatility at increasing temperatures and preservation of physical-chemical state of microemulsion at varying temperatures.”
We also made changes in Fig. 3 and colored schematic aggregates.
Reviewer 2 Report
Comments and Suggestions for Authors
The relaxation behavior of water confined in reverse micelles in water/sodium bis(2-ethylhexyl) sulfosuccinate/decane system was studied by DLS, NMR, and dielectric measurements. “Wait and switch” relaxation mechanism was shown to well describe the studied case. The manuscript is generally well-written and can be suitable for publication after a minor revision.
1) AOT abbreviation should be clarified (sodium bis(2-ethylhexyl) sulfosuccinate) upon first usage as IJMS is a mulitdisciplinary journal.
2) Terms "dielectric measurements" and "conductometry" are mixed in the manuscript, which should be avoided.
3) Color legend is missing for Figure 10
Author Response
Dear Reviewer,
Thank you for valuable comments and recommendations.
- We added the AOT chemical name to the Section 2.1. Structural Features of AOT-based Water-in-Oil Microemulsions.
- We reread our text attentively and tried to correct misunderstanding in the text on using terms "dielectric measurements" and "conductometry". In ani case, it is necessary to take in consideration that these two phenomena are closely bound for conducting dielectrics. In Fig. 6 one can see that dielectric spectra consist from relaxation processes and conductivity contribution.
- We added color legend to Figure 10.
Reviewer 3 Report
Comments and Suggestions for Authors
I am positive for publishing this manuscript, after revising few issues within the text. Respected Authors may find my comments as follows:
1- Page 1, line 36: The first sentence needs to rewrite in more scientific way. For instance Authors may start with "since the first experimental measurements of dielectric constant of water at liquid sate.....
2- Page 1m line 42: The unique properties of water such as....
3- Page 3, line 105 : H1NMR-PFG, please add to section 3.3 as well.
4- Page 4, line 122: ""In our opinion " ---> We anticipate
5- Page 5, line 154: Please remove the word "Obvious".
6- Page 5, line 161, Fig 3 caption: short description then (A) Electric conductivity. (B)...
7- Page 5, line 169: H1NMR, please check and correct this throughout the txt.
8-Page 6, Figure 4: same as point 6
9-Page 8, Figure 6, caption: Experimental data (black squares). Please add clear explanatory txt to panel A as well. Please add enough explanatory txt about symbols and color codes to all figures for the readers.
10- Page 8, lines 252 and 254: 16 kJ.mol-1 and 15.9 kJ.mol-1
11- Page 11, line 313, Fig 9 caption: Missing explanation on panels of figures, what are those blue circles representing?
Author Response
Dear Reviewer,
Thank you for valuable comments and recommendations. We took them into account and made corrections in blue in pdf format of revised article:
1- Page 1, line 36: The first sentence needs to rewrite in more scientific way. For instance Authors may start with "since the first experimental measurements of dielectric constant of water at liquid sate.....
Answer: We agree with you. We made corrections according to your preposition.
2- Page 1m line 42: The unique properties of water such as....
Answer: Corrected.
3- Page 3, line 105 : H1NMR-PFG, please add to section 3.3 as well.
Answer: Corrected.
4- Page 4, line 122: ""In our opinion " ---> We anticipate
Answer: Corrected.
5- Page 5, line 154: Please remove the word "Obvious".
Answer: Corrected.
6- Page 5, line 161, Fig 3 caption: short description then (A) Electric conductivity. (B)...
Answer: Corrected.
7- Page 5, line 169: H1NMR, please check and correct this throughout the txt.
Answer; Corrected.
8-Page 6, Figure 4: same as point 6
Answer: Corrected.
9-Page 8, Figure 6, caption: Experimental data (black squares). Please add clear explanatory txt to panel A as well. Please add enough explanatory txt about symbols and color codes to all figures for the readers.
Answer: It is not necessary to use A, B and C to name the panels, because they are named according to frequency ranges – LF, MW, HF. The explanation of these abbreviations are present in the text.
10- Page 8, lines 252 and 254: 16 kJ.mol-1 and 15.9 kJ.mol-1
Answer: We don’t see mistake here.
11- Page 11, line 313, Fig 9 caption: Missing explanation on panels of figures, what are those blue circles representing?
Answer: Done.
Reviewer 4 Report
Comments and Suggestions for Authors
This is a thorough and well-documented study on the relaxation behavior of water confined in reverse micelles. The scientific background is well presented and the conclusions answer to the scientific questioning. The contribution will most probably deserve acceptation after completion/clarification of the text regarding the following points.
- Equation 1 and line 102. Actually, the references [30,31] only estimate the size of the water pool R_W = 0.15 W. To estimate the size of the entire micelle R_M, authors add to R_W the length of the stretched AOT molecule L_AOT. This is not realistic since the model considers spherical micelles for which the available area per AOT molecule increases with the distance to R_W. It would have been more appropriate to estimate R_M from a AOT hollow sphere calculated from W ratio and AOT volume. In any case, it is essential that authors discuss in the manuscript their choice of second term for equation 1.
- Line 107: "the origins of these differences are still unclear". This is too short: authors have to give their hypotheses on the origin(s) of these differences, all the more as the significance of their own results is affected. As a matter of fact, all these methods do not measure the same objects. For instance, DLS measures the hydrodynamic radius i.e. the object formed by the water pool, the AOT shell and a thin shell of solvent moving with the micelle whereas SAXS measures the radius of the water pool. Additionally, AOT micelles are polydisperse, and methods give different mean radii depending on how much they overweight/underweight the large and small micelle fractions.
- Figure 3 and text. Authors show the size distribution by volume and number, so please calculate the polydispersity and comment it in the text. Polydispersity is an important characteristic of the system that should be specified and commented in the manuscript.
- Lines 116-117. Please, propose explanations for this deviation. Beyond W=15, it is obvious that R_M stops increasing with W and becomes more and more polydisperse. A likely interpretation is that the expansion of the water pool continues with a micelle shape changing from spherical to elongated. In this case, the Stokes−Einstein equation is not any more valid and DSL measures an apparent radius which is not far from the small radius of the real elongated object. Of course, elongated micelles are intrinsically polydisperse in size and shape, agreeing with the experimental trend.
- Lines 117-119. As commented in my point 2, such a difference of slope is normal since authors measured the hydrodynamic radius, which is larger than the micelle radius calculated from equation 1.
- Lines 120-121: "pushing aside water and surfactant molecules". It is not clear what author mean here. This needs to add some explanations and a drawing to the manuscript.
- Additional scheme. Future readers of the same specialized research field will be familiar with AOT. However, IJMS targets a broad readership. So please, insert a figure with the developed chemical formula of AOT.
Author Response
Dear Reviewer,
Thank you for valuable comments and recommendations. We took them into account and made corrections, if necessary, in blue in pdf format of revised article:
- Equation 1 and line 102. Actually, the references [30,31] only estimate the size of the water pool R_W = 0.15 W. To estimate the size of the entire micelle R_M, authors add to R_W the length of the stretched AOT molecule L_AOT. This is not realistic since the model considers spherical micelles for which the available area per AOT molecule increases with the distance to R_W. It would have been more appropriate to estimate R_M from a AOT hollow sphere calculated from W ratio and AOT volume. In any case, it is essential that authors discuss in the manuscript their choice of second term for equation 1.
Answer: We agree with dear Reviewer that all the models which try to give strict equations to estimate real dimensions and size of molecular complexes is always rather relative and cannot be absolutely correct. However, we have used the most popular and prevalent model of AOT based reverse micelle. It is used for estimation of AOT reverse micelles in most of manifold references. We used it in our work as a most accepted.
- Line 107: "the origins of these differences are still unclear". This is too short: authors have to give their hypotheses on the origin(s) of these differences, all the more as the significance of their own results is affected. As a matter of fact, all these methods do not measure the same objects. For instance, DLS measures the hydrodynamic radius i.e. the object formed by the water pool, the AOT shell and a thin shell of solvent moving with the micelle whereas SAXS measures the radius of the water pool. Additionally, AOT micelles are polydisperse, and methods give different mean radii depending on how much they overweight/underweight the large and small micelle fractions.
Answer: This is not the goal of the present work to discuss possible differences in precise estimation of micelle size from different experimental methods, it is quite enough for us to realize that the radius of water pool of AOT-based reverse micelle is roughly proportional to W, varying in our study about one order of magnitude or three orders in the volume, influencing significantly the water properties. We have added this fragment to our text to clarify our goal.
- Figure 3 and text. Authors show the size distribution by volume and number, so please calculate the polydispersity and comment it in the text. Polydispersity is an important characteristic of the system that should be specified and commented in the manuscript.
Answer: Thank you for this comment. We made corresponding proofs to Fig.2 and added the size distribution for DLS results in Fig. 2 – every data point for size distribution by volume and by number is the average of 5 measurements with corresponding distribution of obtained results.
- Lines 116-117. Please, propose explanations for this deviation. Beyond W=15, it is obvious that R_M stops increasing with W and becomes more and more polydisperse. A likely interpretation is that the expansion of the water pool continues with a micelle shape changing from spherical to elongated. In this case, the Stokes−Einstein equation is not any more valid and DSL measures an apparent radius which is not far from the small radius of the real elongated object. Of course, elongated micelles are intrinsically polydisperse in size and shape, agreeing with the experimental trend.
Answer: Dear Reviewer, your supposition and proposition are indeed very interesting. But too complicated. We suppose that our interpretation and explanation are well enough for this article. We’ll try to use your helpful offer in our future work.
- Lines 117-119. As commented in my point 2, such a difference of slope is normal since authors measured the hydrodynamic radius, which is larger than the micelle radius calculated from equation 1.
Answer: We suppose that your preposition is rather interesting, but now we don’t have possibility to complicate our article and change the proposed interpretation.
- Lines 120-121: "pushing aside water and surfactant molecules". It is not clear what author mean here. This needs to add some explanations and a drawing to the manuscript.
Answer: We made changes to this place and simplify the text.
- Additional scheme. Future readers of the same specialized research field will be familiar with AOT. However, IJMS targets a broad readership. So please, insert a figure with the developed chemical formula of AOT.
Answer: We took into account this comment and insert AOT formula in Fig. 5.
Round 2
Reviewer 4 Report
Comments and Suggestions for Authors
Authors made the necessary revisions. The manuscript can be accepted in present form.